# WRSS: An Object-Oriented R Package for Large-Scale Water Resources Operation

**Rezgar Arabzadeh** [1], **Parisa Aberi** [1], **Sina Hesarkazzazi** [1], **Mohsen Hajibabaei** [1], **Wolfgang Rauch** [1], **Saman Nikmehr** [2,*] and **Robert Sitzenfrei** [1]

[1]  Unit of Environmental Engineering, Department of Infrastructure Engineering, Faculty of Engineering Sciences, University of Innsbruck, Technikerstrasse 13, 6020 Innsbruck, Austria; rezgar.arabzadeh@uibk.ac.at (R.A.); parisa.aberi@uibk.ac.at (P.A.); sina.hesarkazzazi@uibk.ac.at (S.H.); mohsen.hajibabaei@uibk.ac.at (M.H.); wolfgang.rauch@uibk.ac.at (W.R.); robert.sitzenfrei@uibk.ac.at (R.S.)

[2]  Department of Water Science and Engineering, University of Kurdistan, Sanandaj 66177-15175, Iran

*  Correspondence: s.nikmehr@uok.ac.ir

**Abstract:** Water resources systems, as facilities for storing water and supplying demands, have been critically important due to their operational requirements. This paper presents the applications of an R package in a large-scale water resources operation. The *WRSS* (Water Resources System Simulator) is an object-oriented open-source package for the modeling and simulation of water resources systems based on Standard Operation Policy (SOP). The package provides R users several functions and methods to build water supply and energy models, manipulate their components, create scenarios, and publish and visualize the results. *WRSS* is capable of incorporating various components of a complex supply–demand system, including numerous reservoirs, aquifers, diversions, rivers, junctions, and demand nodes, as well as hydropower analysis, which have not been presented in any other R packages. For the *WRSS*'s development, a novel coding system was devised, allowing the water resources components to interact with one another by transferring the mass in terms of seepage, leakage, spillage, and return-flow. With regard to the running time, as a key factor in complex models, *WRSS* outshone the existing commercial tools such as the Water Evaluation and Planning System (WEAP) significantly by reducing the processing time by 50 times for a single unit reservoir. Additionally, the *WRSS* was successfully applied to a large-scale water resources system comprising of 5 medium- to large-size dams with 11 demand nodes. The results suggested dams with larger capacity sizes may meet agriculture sector demand but smaller capacities to fulfill environmental water requirement. Additionally, large-scale approach modeling in the operation of one of the studied dams indicated its implication on the reservoirs supply resiliency by increasing 10 percent of inflow compared with single unit operation.

**Keywords:** water resources; R package; standard operating policy (SOP); OOP

## 1. Introduction

Global concerns about water security have increasingly tackled the importance of water management and the operation efficiency of hydro resources. This could be even more critical under limited freshwater resources and growing populations in arid and semi-arid regions where there are frequent and extended periods of water supply deficiency. To make a water resource robust against such anomalies, suitable operational models must be used to incorporate the main functions of the system alongside the involved subsystems. To address this issue, there have been some efforts in the literature, in many of which modeling and simulation have been served as a basis for studying the water system's characteristics and comprehending impacts of governing mechanisms [1–8]. Since most water resources systems are viewed as complex infrastructures, hydrological models emerged to simplify their evaluations and functions [9–14]. In many of the aforementioned studies, computers

are hired to set up the models and evaluate the impact of factors shaping the system functions [15].

Hydrological models have been increasingly used in the water resources systems analysis and simulation. Figure 1a displays a sharp rise in the number of daily released software packages onto one of the most famous and more reliable R project's repositories, also known as Comprehensive R Archive Network (CRAN). Similarly, on the same repository, libraries dedicated to hydrological analyses have experienced an increasing trend in the number of releases between 2006 and 2019 (see Figure 1b). MODIStsp, Evapotranspiration, dynatopmodel, soil physics, rtop, hydroPSO, and ClimDown are examples of the recent open-source developments in this field [16–22].

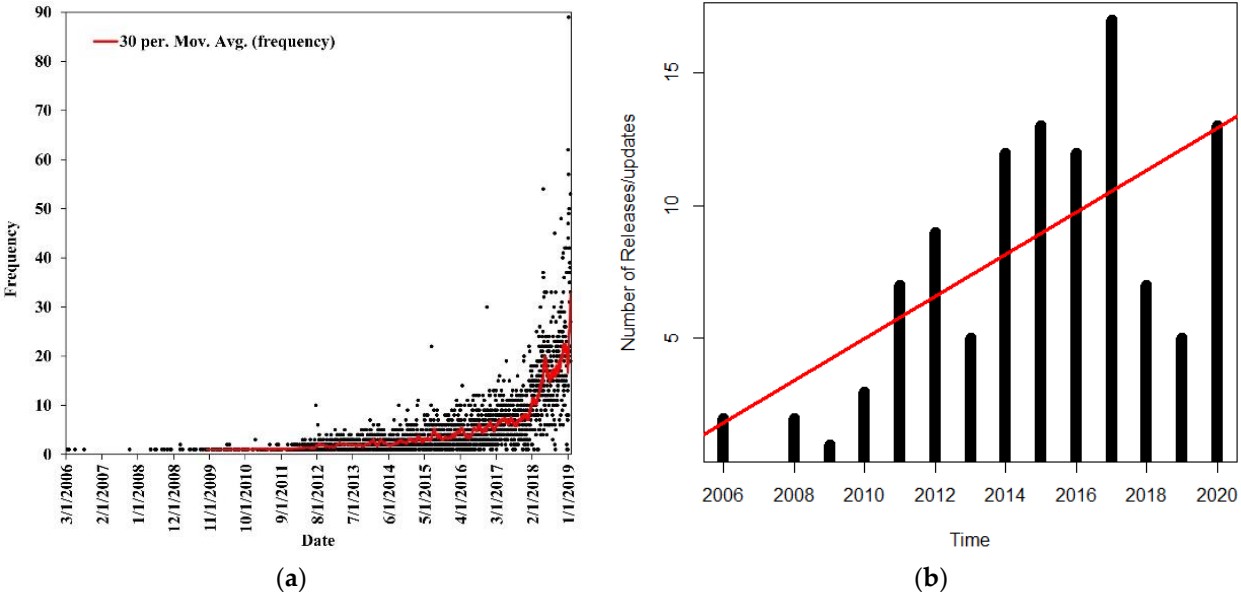

**Figure 1.** (**a**) The number of R packages released on CRAN in the past 14 years (source: R project website: https://cran.r-project.org/web/packages/available_packages_by_date.html (accessed on 18 October 2021)); (**b**) the number of R packages developed for hydro-science and engineering.

For water resource evaluation purposes, there is currently a limited number of simulators, most of which are commercial or non-open source, such as the water evaluation and planning system (WEAP), MODSIM, and HEC-ResSim [9,11,23]. In terms of free and open-source tools, the available options are mostly dedicated to the hydrological cycle and atmospheric studies. For instance, "Evapotranspiration", developed by Guo et al. (2011) [24], estimates station-based potential evapotranspiration using 21 methods, or the "airGR" and "TUWmodel" packages developed by Coron et al. (2017) [25] and Parajka et al. (2013) [26], respectively, are rainfall-runoff simulators. In one case, an R package named as "reservoir", developed by Turner and Galelli (2016) [27], was devised for single-unit reservoir optimization and simulation; however, it is unable to handle large-scale systems. On R's data wrangling rival platform—Python—there have been recent endeavors to develop tools to address different aspects of water resources processes and paradigms such as tracer hydrology, soil moisture retention functions, water economy, hydrological data sharing, as well a water resources systems simulation and optimization [28–32]. Table 1 shows a selected list of available software/libraries dedicated to water resources systems simulation and analysis compared with the R package introduced in this paper. Including *WRSS* 7, commercial/non-commercial software packages have been chosen and compared in terms of execution speed and a selected number of other capabilities. These criteria were categorized into general characteristics, i.e., supporting large-scale model or optimization, hydroelectric, and other capabilities such as execution speed, objects prioritization, and supporting various hydro-infrastructures. The table shows that most of tools are not sup-

porting open-source platforms, except *WRSS*, RSSOP, and reservoir, while many of them are allowing large-scale modeling. In addition, most of the toolkits have possibilities to simulate hydro-electrical energy generation while a few of them are supporting systems optimization. In terms of simulation speed, most of the software packages propose a promising short execution time such as *WRSS* and RSSOP.

**Table 1.** Comparison of water resources tools.

| Software or Package | General Characteristics | | | | | | Hydro-Electric | | | Other Capabilities | | | | | | | |
|---|---|---|---|---|---|---|---|---|---|---|---|---|---|---|---|---|---|
| | Open-Source | Optimization Support | OOP | Rainfall-Runoff Module | Large-Scale | Scripting Support | Hydroelectric | Multiunit Simulation | Penstock-Turbine Sizing | Reservoir | Aquifer | Diversion | Prioritization | User Interface | Execution Time (Simulation) | Execution Time (Optimization) | Hydrologic Mechanisms |
| WRSS | ✓ | × | ✓ | × | ✓ | ✓ | ✓ | ✓ | ✓ | ✓ | ✓ | ✓ | ✓ | × | SF * | × | ✓ |
| WEAP | × | × | ✓ | ✓ | ✓ | ✓ | ✓ | × | ✓ | ✓ | ✓ | ✓ | ✓ | ✓ | M | × | ✓ |
| MODSIM | × | ✓ | ✓ | × | ✓ | × | ✓ | × | ✓ | ✓ | × | × | ✓ | ✓ | M | M | ✓ |
| RSSOP | ✓ | × | × | × | ✓ | ✓ | × | × | × | ✓ | × | × | × | × | SF | × | × |
| HEC-ResSim | × | × | ✓ | × | ✓ | × | ✓ | ✓ | ✓ | ✓ | × | ✓ | ✓ | ✓ | F | × | ✓ |
| reservoir | ✓ | ✓ | × | × | × | ✓ | ✓ | × | × | ✓ | × | × | × | × | F | F | × |
| RIBASIM | × | × | ✓ | × | ✓ | × | ✓ | ✓ | ✓ | ✓ | ✓ | ✓ | ✓ | ✓ | M | × | ✓ |

\* For a single unit reservoir: SF (Super-Fast ~ T), F (Fast ~10 T), M (Medium ~ 20 T), where T is the mean execution time of an SF model on a given computer setup.

The execution speed categories present the running time for a single unit storage reservoir for 10 years of monthly simulation periods. The simulations were conducted on a computer with an Intel (R) Core (TM) i7-4790 CPU (4.00 GHz) CPU and 32 GB installed memory (RAM).

In the present study, R is selected as the platform for the development of a package due to its libraries' ecosystem diversity and richness, through which hydrologists can synergize the results of data processing. In addition, developing open source tools provides scientists and engineers faster features development and bug fixing, quicker and more effective global software product promotion, and, ultimately, technological advancement [33]. In spite of the availability of software out there to analyze and simulate large-scale water resources systems, there is no open-source interface and freely accessible tool to handle the job. In this context, large-scale water resources signify a system comprised of two or more interconnected subsystems. Large-scale water resources systems have been frequently studied by researchers such as cascade reservoirs or complex reservoir-supply networks [34–37]. These systems may interact with each other through hydrological processes including, but not limited to, seepage, leakage, spillage, etc.

Although R packages developed for hydrology and water resources are proposing great advantages, there is no library in R supporting large-scale water resources evaluation. To bridge this gap, an R-based package is introduced in this study to handle large-scale water systems simulation. The package is titled Water Resources System Simulator (*WRSS*), which assists water resources experts in both sizing and operating a complex water resources system.

To handle the complexities raised from large-scale water system operation, *WRSS* uses Object-Oriented Programming (OOP). What makes *WRSS* distinctive from existing developments within R libraries, specifically from the "reservoir" package, is its capability

in multiple hydro-structures operation (e.g., aquifers, reservoirs, etc.) with the possibility of object interaction and prioritization. In contrast to the "reservoir" package that is limited to an isolated reservoir object, *WRSS* supports a range of features capable of interacting with each other through mechanisms such as allocation, return-flow, leakage, seepage, and spillage. Such interplay between features is of high importance due to the complex and interdependent nature of water resources systems [38].

In addition, while energy generation by "reservoir" package is available, there are still some nuances making *WRSS* preferable in practice for hydroelectric simulation. For instance, the energy losses associated with the penstock and the turbine are not considered in "reservoir", while *WRSS* is able to calculate both losses separately. Furthermore, *WRSS* considers extra energy-related specifications for a power plant facility such as turbine-axis-elevation, submergence condition, and tailwater discharge-elevation-function, while these are not included in "reservoir".

This paper aims to review the functionalities of the *WRSS*, an R package dedicated to large-scale water resources systems operation and assessment. The proposed package implementation in R can be of a profound advantage to water resources modelers since there has not yet been developed any other R packages to handle complex water supply–demand networks. Additionally, given *WRSS*'s capabilities and R's growing world of complex computational methods, its users can make more sophisticated models much simpler to address hydro-environmental problems. For instance, tedious and technically challenging tasks such as making a coupled water resources–statistical model goes straight as all of these are available under a unified platform. Therefore, *WRSS* aims to enhance water resources management practices by providing solutions and methods for large-scale systems simulation and analyses.

## 2. Methodologies

### 2.1. Platform Environment

Scientists and engineers concerned with water resources topics have the opportunity to benefit from already developed R packages for water resources studies [39–43]. To implement *WRSS*, R platform was chosen because it is a high-level programming language running on various types of operating systems and easy to understand with widely used, freely available, and trusted packages [44]. R is an environment and a language for graphics and statistical computing. According to only R's main repository (checked on 30 January 2020 (CRAN Packages)), there are 8060 packages, which is far higher than any other data wrangling platforms (e.g., SAS, SPSS, and Python), not to mention R's popularity among both developers and users. An extensive investigation by the Data Science Service [45] indicates that R has a rising trend in popularity, and about half of data science jobs call for R experts. The same blog, additionally, reports R as the second software platform used by scientists in the scholarly articles. In 2009, *The New York Times* ran an article displaying R's growth, and the reason why it is popular among scholars and reports, imposing the threat to commercial packages such as SPSS and SAS [46].

### 2.2. Governing Equations

While there are multiple release policies used for different conditions, a similar procedure to that of Rippl (1883) [47], Standard Operating Policy (SOP), is considered for supplying specific target in *WRSS*. SOP is the simplest and easiest policy used for water evaluation models, which aims to release a quantity of water equal to that needed for water demand, if possible [48].

SOP delivers the claimed demand, if enough water is available, retains extra water before reservoir is full while the target is fully supplied, and surpluses the exceeded water when the temporal capacity exceeds the top storage (see Figure 2a thick line) [49]. As a result, it is an optimal operation method when the reservoir objective is to minimize deficits over the decision space [50].

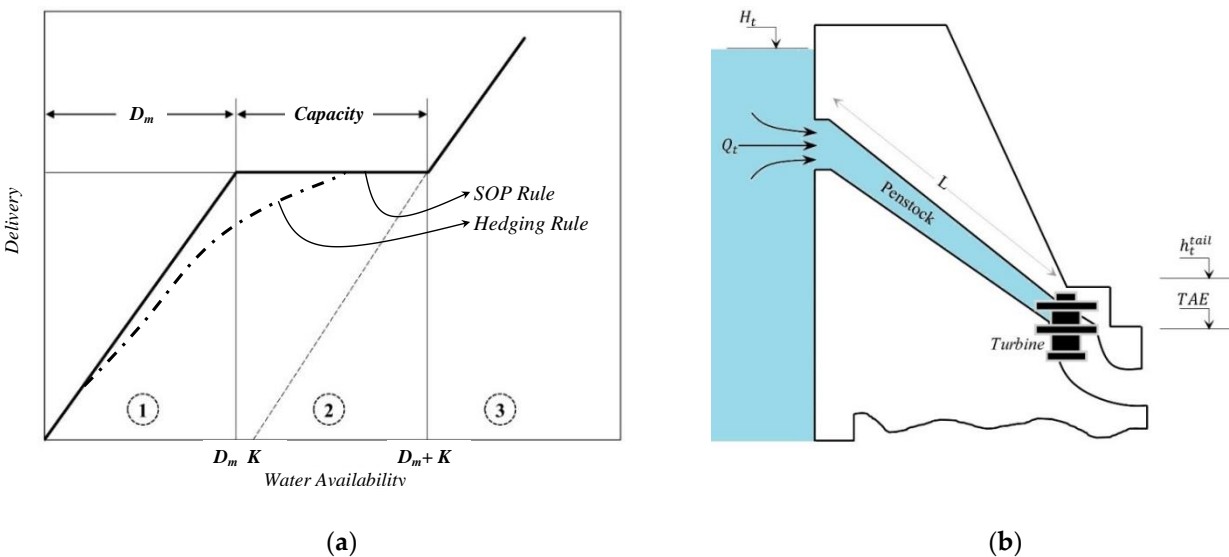

**Figure 2.** (**a**) Standard operating policy for reservoir operation. (**b**) A common impounding facility of hydropower turbine.

In contrast to the SOP, if the delivering amount falls between zero and the demand, it establishes a hedging rule which does not fully satisfy the demands and saves some water for the future (see Figure 2a dashed line). Hedging rules are very useful during long-lasting droughts, which can significantly enhance a system's performance such as reducing vulnerability [51]. Although SOP does not rationalize releases in the future demand, it is practical, easy to understand, and the most often used approach for feasibility studies in development projects [52].

### 2.2.1. Mass Balance

SOP is implemented by mass balance as the core equation for water resources components simulation. Let $k$ be the index of a feature under operation, $i$ be the index of inflow representing the flow released from $i^{th}$ feature to $k^{th}$ feature, and $j$ be the index of outflow released from $k^{th}$ feature to $j^{th}$ feature in $t^{th}$ time step; therefore, the mass balance equation for a component being operated will be as follows:

$$S_{t+1}^k = S_t^k + \sum_i Q_{t,k}^i - \sum_j O_{t,j}^k \tag{1}$$

where the equation represents the system future state based on the current/past state. For a given storage system, $k$, in $t^{th}$ time step, the equation demonstrates that the storage in the future, $S_{t+1}^k$, is equal to the storage in the past, $S_t^k$, plus all inflows, $Q_{t,k}^i$, and all outflows, $O_{t,j}^k$, with a negative sign.

Since features such as junctions, diversions, and reaches are assumed to have a negligible storage capacity ($S$) within two subsequent time steps, Equation (1) can be simplified as follows:

$$\sum_i Q_{t,k}^i = \sum_j O_{t,j}^k \tag{2}$$

where outflow matrix, $O$, can be established from different sources including water supply withdrawals, seepage, or evaporation losses, etc. Equations (1) and (2) are the basis used

for operating all objects available in *WRSS*. For an impounding facility, Equation (1) is rewritten as grouped Equation (3):

$$S_{t+1}^k = S_t^k + \sum_i Q_{t,k}^i - Sp_t^k - \overline{EV}_t^k - \sum_d Re_{t,d}^k - Se_t^k \quad s.t:$$

$$S_{min}^k < S_t^k < S_{max}^k \quad \& \quad \overline{EV}_t^k = \frac{A_t^k + A_{t+1}^k}{2} E_t^k \quad \& \quad Se_t^k = \frac{S_t^k + S_{t+1}^k}{2} \omega^k \tag{3}$$

where it estimates the reservoir's future/current state by subtracting all losses, i.e., evaporation and seepage, from the available water in the current/past state. See the annotation section for terms definition in the equations.

Similarly, for aquifer systems, Equation (1) is rewritten as below:

$$S_{t+1}^k = S_t^k + \sum_i Q_{t,k}^i - \sum_d Re_{t,d}^k - Se_t^k \quad s.t:$$

$$0 < S_t^k < S_{max}^k \quad \& \quad S_{max}^k = V^k \times \varphi^k \quad \& \quad Se_t^k = \frac{S_t^k + S_{t+1}^k}{2} \omega^k \tag{4}$$

where the total available water in the storage is defined as a product of wetted volume and the aquifer-specific storage. Similar to the reservoirs, the seepage volume is defined as the product between the seepage ratio and the average storage between two subsequent time steps as follows:

$$\delta_t^k = \sum_i Q_{t,k}^i - Ex_t^k - RV_t^k \quad s.t: \; Ex_t^k = max\left(0, \sum_i Q_{t,k}^i - D_t^k\right) \quad \& \quad RV_t^k = \delta_t^k \times RF^k \tag{5}$$

Equation (5) represents the mass balance in a demand node where, for every temporal step, an effective supplied water is calculated as the difference between the total inflows and excess/return flows, where the excess flow is calculated as the difference between the total inflow and demand, while the return flow is defined as a linear function of return flow fraction and the effective supplied water.

Assuming negligible storage/losses for a diversion facility, Equation (1) will be simplified as Equation (6), where the system outflow is computed as the difference between the total inflow and the diverted volume:

$$O_t^k = \sum_i Q_{t,k}^i - Dv_t^k \quad s.t: \; Dv_t^k = min\left(Cap^k, \sum_i Q_{t,k}^i\right) \tag{6}$$

To simulate rivers/channels system outflow, where applicable, all losses and seepages are subtracted from inflows. The seepage is computed as a fraction of total inflows:

$$O_t^k = \sum_q Q_{t,q}^k - Re_t^k - Se_t^k \quad s.t: \; Se_t^k = \omega^k \sum_i Q_{t,k}^i \tag{7}$$

In *WRSS*, similar to diversion facilities, losses and storage are assumed to be negligible in junctions, so Equation (1) simplified as below:

$$O_t^k = \sum_i Q_{t,k}^i \tag{8}$$

where the outflow is set to be equal to the inflow.

### 2.2.2. Objects Prioritization

To incorporate targets/resources *supplying/operation* priorities, an integer value in [1, 99] interval was defined for every feature, presenting *allocation/operation* superiority, where the smaller value is translated to a higher allocation/operation order and vice-versa. To consider objects interactions, a method was developed to detect priorities of not only basin

features simulation from upstream to downstream but also supply and demand operation. Accordingly, let $\vec{\alpha_i}$ be a vector of objects unique numbers, $\vec{\beta_i}$ be $\vec{\alpha_i}$'s downstream objects unique number, and $\vec{\gamma_i}$ be a vector of priorities corresponding to the $\vec{\alpha_i}$, $k$ group (s), $g$ of the object (s) in the same level of simulation priority could be established as follows:

$$g^k = \{\forall\ \alpha_i \neq \beta_j | i,j \in \{1,2,\ldots,|\alpha|\}\} \ \ \& \ \ g^{k+1} = \{\forall\ \alpha_i \neq \beta_j | i,j \in \{1,2,\ldots,|\alpha| \cap g_k'\}\} \tag{9}$$

$$\forall\ 1 \leq\ i \leq N\ \left|\left\{i \in N \middle| g_a^k = g_i^k\right\}\right| = \left|\left\{i \in N \middle| \gamma_a = g_i^k\right\}\right| \tag{10}$$

Equation (9) detects and groups objects from upstream to downstream; then, using the priorities given in $\vec{\gamma}$, the objects within $g^k$ are sorted in ascending order. To control the algorithm flow, it is assumed that all targets and objects recharging from external source(s) are located downstream of their corresponding supplier(s)/recharger(s). The following pseudo-code (see Algorithm 1) represents the mathematical approach described above:

---

**Algorithm 1**

---

 Populate a reference matrix code whose columns correspond to objects and rows are attributes of the objects as follows:
      1- label 2- downstream label 3- priority
**Loop**
      Check which label(s) in the first row of reference matrix is/are not duplicated in the second row and select them as upstream feature(s)
      **Loop**
         Select a feature from the upstream set with higher priority as *current_object*
            **If** the *current_object* is a water resource, then:
               Simulate the feature and allocate water to demand site(s) supplied by *current_object* according to their priority(ies))
               Route the outflows to the downstream of the current_object
          **End If**
            **If** the *current_object* is a demand site:
               Compute the return-flow fraction volume and route it to the downstream of the *current_object*
          **End If**
            Terminate the loop if the criterion (number of iterations > the number of upstream feature(s)) is met
      **End Loop**
      Remove upstream features from the reference matrix
      Terminate the loop if the criterion (number of columns in reference matrix is zero) is met
**End Loop**

---

The algorithm detects objects from upstream to downstream. Then, those objects in the most upstream location and with the highest priority are selected for operation (*current_object*). If the *current_object* is a water resource, then the algorithm simulates the feature and allocates water to demand site (s) connected to the *current_object* according to their priority (ies) then routes the outflows to the downstream. If the *current_object* is a demand node, algorithm calculates return-flow fraction volume, where applicable, and routes it to the downstream. The process is performed until all objects in the model are simulated at least once.

For shared water resources supplying multiple targets with equal priority, the allocation is conducted based on each demand's volume. Let $Re_t^k$ be the released volume from the $k^{th}$ resources in $t^{th}$ time step and $\left\{De_t^1, De_t^2, \ldots, De_t^d\right\}$ be the target values, all with equal priority being supplied by the $k^{th}$ resources, the allocation for each target, $Re_{t,d}^k$ is calculated as below:

$$Re_{t,d}^k = \frac{De_t^d}{\sum_d De_t^d} Re_t^k \tag{11}$$

### 2.2.3. Hydroelectric Energy Generation

Hydropower energy generation has been implemented in *WRSS* version 2.0 and above, however, it is limited to reservoirs. The most common type of hydroelectric power plant is an impoundment facility in which water is released from the reservoir by a large pipe known as "penstock", flowing through a turbine, spinning it, which in turn activates a generator to produce electricity (see Figure 2b). The following equation calculates the energy generated by a power plant:

$$P_t = \rho g Q_t H_t \varphi_t \quad s.t:$$
$$H = \frac{H_t^{(1)} + H_t^{(2)}}{2} - h_t^{tail} - hf_t$$
$$h_t^{tail} = \left\{ \begin{array}{cc} max\left(TAE, \, h_t^{tw}\right) & submerged \\ TAE & !submerged \end{array} \right\} \tag{12}$$
$$= hf_t^T + hf_t^P = hf_f^T + 10.67 \frac{L}{D^{4.804}} \left( \frac{Q_t}{C} \right)^{1.852}$$
$$\varphi_t = \varnothing \left( Q_t \right)$$

In Equation (12), there are two terms with unknown values, $Q_t$ and $H_t^2$, needed to be determined using trial and error procedure. First, an assumption of release value is considered; then, $H_t^2$ and $P_t$ are calculated. Next, the constraints are checked, and the procedure is repeated until an insignificant change between the generated power and installed capacity is observed. The following equation represents the trial and error procedure as an optimization problem:

$$min\{|P_t - P_{Installed}|\} \quad s.t:$$
$$P_t \quad < P_{Installed}$$
$$\left\{ H_t^1, H_t^2 \right\} \quad \in [min(DH), \ max(DH)] \tag{13}$$
$$Q_t \quad \in [min(DQ), \quad max(DQ)]$$

To solve Equation (13), *WRSS* uses the Improved Stochastic Ranking Evolution Strategy optimization algorithm, whose details can be found in [53].

### 2.2.4. Performance Indices

The performance of a water resources system is defined as its ability to meet the downstream requirements and, if possible, store water for future. Performance indices are categorized into yield-based and risk-based approaches (refer to [54]). *WRSS* uses the risk-based approach, implemented in the *risk* function, which includes measures known as reliability, vulnerability, and resiliency [50]. The measure formulations and definitions are as below:

The probability of a reservoir to release water ($Re_{t,d}^k$) equal to $D_t^k$ is defined as *reliability*, which can be defined in both temporal and volumetric scales. The temporal method considers the total number of periods $Re_{t,d}^k$, which meets the $D_t^k$ by a defined threshold ($\alpha$). The volumetric reliability is the same as temporal reliability in which the fluid volumes are considered instead. The reliability indices for both temporal and volumetric criteria are presented in Equations (14) and (15), respectively:

$$rel_{tmp} = \frac{N\left(Re_{t,d}^k > \alpha.D_t^k\right)}{T} \qquad \forall : t \in [1\ldots, T] \qquad (14)$$

$$rel_{vol} = \frac{\sum_{t=1}^T \left(\left|Re_{t,d}^k < \alpha.D_t^k \ \& \ \alpha.D_t^k \middle| Re_{t,d}^k \geq \alpha.D_t^k\right.\right)}{T.\alpha.D_t^k} \qquad \forall : t \in [1\ldots, T] \qquad (15)$$

The speed that a hydro-system is recovered from a temporal failure is defined as *resiliency*. The following is the resiliency mathematical expression:

$$res = \frac{N\left(Re_{t,d}^k < \alpha.D_t^k \middle| Re_{t+1,d}^k \geq \alpha.D_t^k\right)}{T.\alpha.D_t^k} \qquad \forall : t \in [1\ldots, T] \qquad (16)$$

To measure the magnitude of failures, the vulnerability index is used as expressed in Equation (17):

$$vul = \frac{\sum_t \left(\alpha.D_t^k - Re_{t,d}^k \middle| Re_{t,d}^k < \alpha.D_t^k, 0 \middle| Re_{t,d}^k \geq \alpha.D_t^k\right)}{T} \qquad \forall : t \in [1\ldots, T] \qquad (17)$$

### 2.3. Package Skeleton

*WRSS* uses OOP concepts to simulate any given project layouts. OOP simplifies processes through some definitions known as "*objects*", designed in such a way that they can interact with one another. Figure 3 represents objects, classes, and methods implemented in *WRSS* shown by Unified Modelling Language (UML). As shown in Figure 3, the package includes six classes for feature construction, namely: *createRiver*, *createReservoir*, *createAquifer*, *createJunction*, *createDiversion*, and *createDemandSite*. For instance, the class of *createArea* signifies the basin object construction, in which, *addObjectToArea* method adds all constructed features to the basin. Consequently, basin object could be passed to *sim* class where *plot* and *risk* methods for the simulation, visualizations, and extra analyses can be performed, respectively.

The classes and functions of *WRSS* have been categorized into three groups, including object manipulation, analysis, and visualization as presented in Table 2, where its second column is the name of the method/class, the third column briefly describes the method/class output, i.e., *reservoirRouting* routes available water through a impounding facility, and the last column describes the main specifications of the methods/classes, i.e., *ripple* method computes the storage capacity based on SPA algorithm. To simulate an object from the class of *createArea*, users should mind adding time series in an appropriate temporal scale. As stipulated in the package user manual (https://cran.r-project.org/web/packages/WRSS/WRSS.pdf (accessed on 18 August 2021)), users can either coerce an optional time-series or leave (a) constant(s) to be cycled throughout the simulation time-scale.

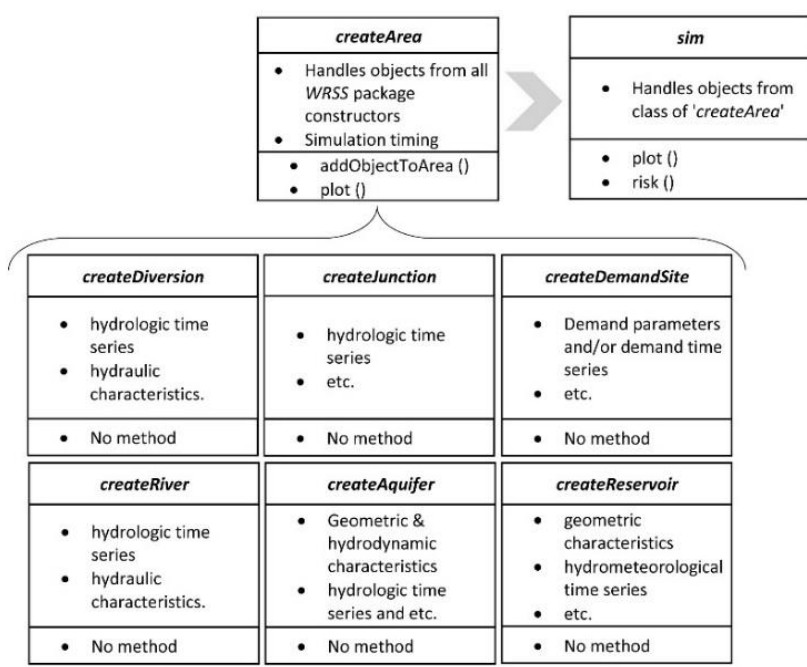

**Figure 3.** Representation of classes and methods built-in *WRSS* package through Unified Modelling Language (UML).

**Table 2.** *WRSS* package functions.

| Category | Classes/Methods | Objective | Specification |
|---|---|---|---|
| objects manipulation | *Constructors* | | |
| | *createArea* | Creates a basin | Requires the number of time steps and intervals length of simulation |
| | *createJunction* | Creates a junction object | Combines flows drained to the junction |
| | *createRiver* | Creates a river or channel object | River with possibility for allocation and seepage |
| | *createReservoir* | Creates a storage reservoir object | Handles reservoir geometry for accurate estimation of evaporation volume |
| | *createDiversion* | Creates a diversion object | A diversion work with a fix diversion rate |
| | *createAquifer* | Creates an aquifer object | Constructs an unconfined aquifer object with a given hydrodynamic parameter |
| | *createDemandSite* | Creates a demand object | Accepts either demand time series or demand parameters |
| | *addObjectToArea* | Add objects inherited from the constructors to an object from class of *createArea* | Manages objects inherited from the constructors and adds them to an object from class of *createArea* |

**Table 2.** *Cont.*

| Category | Classes/Methods | Objective | Specification |
|---|---|---|---|
| Simulation | *sim* | Operates water resources system using standard operation policy on an object inherited from class of *createArea* | Performs standard operation policy for connected reservoirs system |
| | *riverRouting* | Routes flow in a channel or river | Allocates resources to multiple demand sites with priorities |
| | *reservoirRouting* | Routes flow in a storage/hydropower reservoir | Allocates resources to multiple demand sites with priorities |
| | *aquiferRouting* | Routes storage in an unconfined aquifer | Allocates resources to multiple demand sites with priorities |
| | *diversionRouting* | Routes flow in diversion works | Allocates resources to multiple demand sites with priorities |
| | *ripple* | No-fail storage size using Rippl's method | Uses reverse SPA |
| | *cap_design* | Reservoir capacity design | Uses yield–storage relationships |
| Performance analysis and visualization | *risk* | Reservoir performance indices | Reliability, resiliency, and vulnerability |
| | *plot.createArea* | Plot function for object inherited from class of *createArea* | The function uses network analysis to layouts features existing in the basin |
| | *plot.sim* | Plot function for object inherited from class of *sim* | Plots releases, spills, and storages time series |

*2.4. Storage Design*

According to [47], the minimum capacity of a reservoir is an amount of storage required to meet the specified water demand (s) without leading to water supply shortage, which is commonly computed by the 'sequent-peak-algorithm' method [55]. In addition, the reservoir capacity could be designed by the concepts of yield–storage relationships [56]. In this method, the capacity design process includes the calculation of reservoir-yield-based criteria for multiple pairs of capacity and design parameters. For capacity design, *WRSS* proposes the *rippl* function to handle Rippl's method and the *cap_design* for yield–storage relationships.

To determine the size of the reservoirs in the yield-capacity method, since the capacity size is not only a function of dam site river flow but also depends on other project measures (e.g., water demand volume). So, combinations of a number of reservoir's size candidates, i.e., $\overrightarrow{Cap_{max}} = \{Cap_1, Cap_2, \ldots, Cap_m\}$ and any project design parameters, e.g., developable agricultural area, $\overrightarrow{D} = \left\{ \overrightarrow{D_1}, \overrightarrow{D_2}, \ldots, \overrightarrow{D_n} \right\}$ are used to measure any RRV (reliability-resiliency-vulnerability) metrics. For instance, if one design parameter denotes candidates of developable cropland areas, $\overrightarrow{Crop} = \{A_1, A_2, \ldots, A_o\}$, and the other one is $\overrightarrow{Cap_{max}}$, the payoffs would be three RRV values corresponding to any given cropland and capacity size. While each payoff may result in individual capacity/cropland size, Loucks (1997) [57] has proposed a multi-index technique to amalgamate all measures as below:

$$SI_j^i = reliability_j^i \times resiliency_j^i \times \left( 1 - \frac{vulnerability_j^i}{max\left\{ vulnerability_j^i \right\}} \right) \tag{18}$$

where $i$ is the capacity index and $j$ is the cropland area index. The resulting product, SI, ranges from 0, as the lowest and worst possible value, to 1, as the highest and best possible value. This SI applies to each criterion $C$ for any constant level of probability p, further calculated for each alternative system or decision being considered [57]. If there are multiple targets (domestic, agriculture, etc.), a combined weighted relative SI (RSI) as recommended by Loucks (1997) will be utilized as shown below [57]:

$$RSI = \sum_i W_i \times SI_i \qquad (19)$$

where $W_i$ is the relative weight ranging from 0 to 1 and summing 1, which can be defined to reflect the importance of each sustainability index. To determine $W_i$, either an optimization approach or a multi-criteria-decision-making (MCDM) method is recommended. In this study, Analytical Hierarchal Process (AHP) method is used to derive the weights. A detailed explanation about the AHP method could be found in Vaidya and Kumar (2006) [58].

### 2.5. Restrictions and Precautions

*WRSS* uses R version 3.0 or later, and it is dependent on built-in functions *Hmisc, network, nloptr, ggplot2,* and *GGally* packages. Furthermore, it imports *graphics* and *stats* packages, all available when the R core is installed. In addition, *WRSS* supports merely SOP without possessing the capability to incorporate user-defined operation rules or optimization approaches. The current version of *WRSS* supports daily, weekly, and monthly simulation time intervals, and consequently, other time scales are not implemented/integrated yet. Accordingly, precautions must be made by users to avoid invalidity of governing equations for simulation of systems of small units with fine temporal resolutions, i.e., daily, due to dominance of hydraulic processes over hydrological and mass balance equations.

## 3. Case Study

To test *WRSS*, the Zerrine-rud basin in Iran is selected for both operation and design purposes. In the first part, the Bukan dam capacity is estimated by different approaches, and in the second part, a large-scale system of 5 reservoirs and 11 water demands are simulated. Figure 4a,b present a topographic layout and schematic view of the main drainage network, respectively. As seen in Figure 4b, every feature is labeled with numbers from one to three, indicating higher to lower priorities, respectively, and return-flow links are shown by connected dashed lines. The reservoir system is composed of four parallel dams located upstream of the Bukan reservoir, constructed at the west of Iran, located in Kurdistan Province. The region area is about 4700 km$^2$, receiving 510 mm/year rainfall with an annual average temperature of 9.2 °C.

### Datasets

The datasets used for the reservoirs' simulation include hydro-meteorological, demand time series, and reservoirs geometric specifications. All of the datasets are provided by the Iran Water Resources Management Company [59]. Table 3 shows the monthly averaged time series used for the system simulation. The most recent 10 years of monthly time series of hydrologic parameters were used as the representative time series for both dry and wet periods of hydrologic cycle. According to Table 3, about 89% of the basin's surface water goes to the Bukan dam with a capacity about $670 \times 10^6$ m$^3$. Regarding the demands, A1 and D1 are the largest demand nodes supplied by Bukan with an annual water demand about to $1250 \times 10^6$ m$^3$. Generally, the agricultural sector is the main water user in the catchment, and its demands start from May and end in September, while other sectors' water demands are fairly constant with less variation during the year. The evaporation is estimated to be higher as the site location is in higher latitude, and the basin average evaporation is about 1350 mm/year.

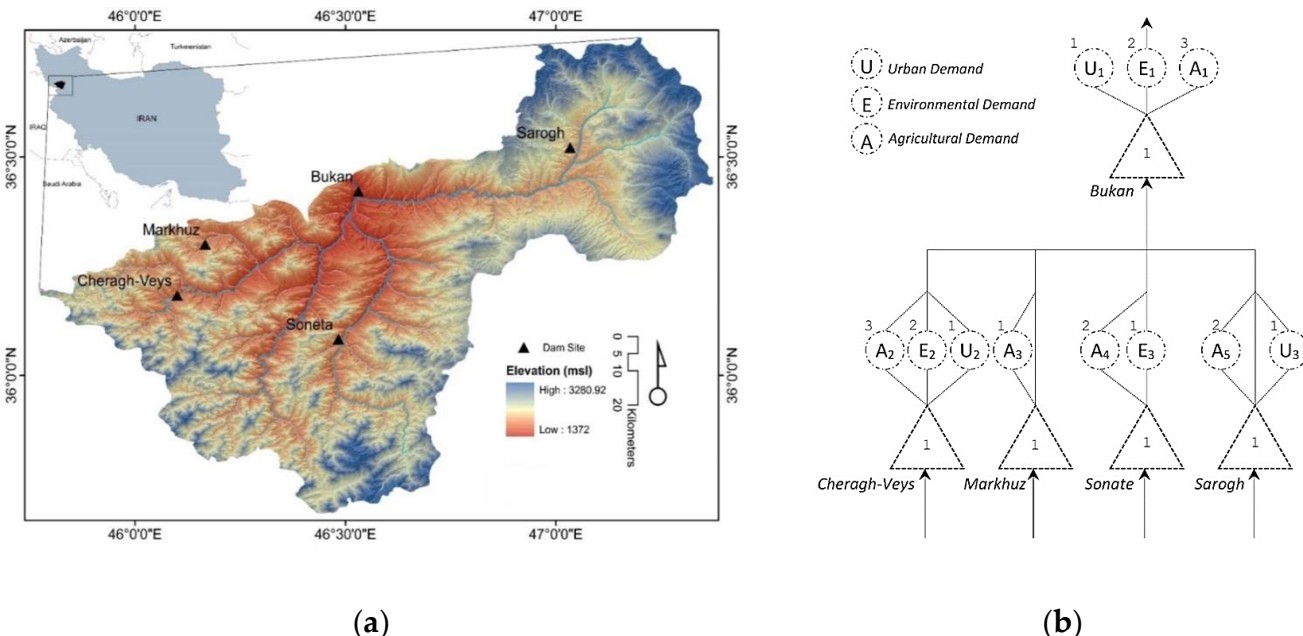

**Figure 4.** (**a**) Locations of dams within the Zerrineh-rud sub-basin in Kurdistan, Iran. (**b**) Schematic model of the Zerrineh-rud features.

**Table 3.** Monthly average time series used in multi-reservoir simulation.

| | | Sep. | Oct. | Nov. | Dec. | Jan. | Feb. | Mar. | Apr. | May. | Jun. | Jul. | Aug. |
|---|---|---|---|---|---|---|---|---|---|---|---|---|---|
| Inflow ($\times 10^6$ m$^3$) | Bukan | 16.49 | 44.97 | 71.82 | 83.14 | 113.39 | 253.08 | 463.12 | 370.66 | 108.85 | 33.64 | 19.42 | 16.18 |
| | Cheragh-Veys | 0.94 | 2.38 | 3.04 | 4.25 | 5.68 | 8.69 | 16.25 | 10.96 | 3.43 | 1.32 | 0.98 | 0.91 |
| | Markhuz | 0.06 | 0.17 | 0.23 | 0.30 | 0.39 | 0.83 | 1.43 | 1.03 | 0.28 | 0.08 | 0.06 | 0.06 |
| | Sonate | 0.43 | 1.33 | 2.17 | 2.67 | 3.95 | 8.94 | 16.66 | 13.57 | 4.59 | 1.20 | 0.59 | 0.42 |
| | Sarogh | 2.34 | 3.47 | 3.24 | 3.86 | 4.51 | 7.11 | 14.69 | 15.61 | 6.66 | 3.26 | 2.34 | 2.12 |
| Demand Sites ($\times 10^6$ m$^3$) * | A1 | 88.00 | 0.00 | 0.00 | 0.00 | 0.00 | 0.00 | 22.00 | 102.00 | 229.00 | 245.00 | 226.00 | 179.00 |
| | A2 | 1.51 | 0.33 | 0.00 | 0.00 | 0.00 | 0.00 | 0.00 | 2.00 | 7.30 | 9.87 | 7.60 | 4.10 |
| | A3 | 0.37 | 0.00 | 0.00 | 0.00 | 0.00 | 0.00 | 0.21 | 1.50 | 2.93 | 2.93 | 2.03 | 1.09 |
| | A4 | 1.54 | 14.48 | 36.92 | 44.23 | 19.56 | 3.17 | 6.22 | 0.40 | 0.00 | 0.00 | 0.09 | 0.22 |
| | A5 | 2.00 | 1.10 | 0.80 | 0.50 | 0.50 | 0.00 | 2.30 | 4.50 | 7.30 | 9.00 | 7.60 | 4.10 |
| | E1 | 1.47 | 4.63 | 7.84 | 8.97 | 11.50 | 75.83 | 150.61 | 117.80 | 11.25 | 3.27 | 1.71 | 1.38 |
| | E2 | 0.31 | 0.31 | 0.31 | 0.31 | 0.31 | 0.30 | 0.32 | 0.32 | 0.32 | 0.32 | 0.32 | 0.32 |
| | E3 | 0.02 | 0.01 | 0.00 | 0.00 | 0.00 | 0.00 | 0.00 | 0.00 | 0.00 | 0.00 | 0.00 | 0.01 |
| | D1 | 14.20 | 12.20 | 12.70 | 13.27 | 13.01 | 13.41 | 11.10 | 13.01 | 13.40 | 14.03 | 14.50 | 13.20 |
| | D2 | 3.61 | 3.41 | 3.20 | 2.78 | 2.99 | 3.16 | 3.68 | 3.85 | 4.15 | 4.29 | 4.41 | 3.93 |
| | D3 | 0.86 | 0.86 | 0.86 | 0.55 | 0.55 | 0.55 | 0.86 | 0.86 | 0.86 | 1.15 | 1.15 | 1.15 |
| Evaporation (m) | Bukan | 0.15 | 0.08 | 0.06 | 0.04 | 0.04 | 0.06 | 0.08 | 0.13 | 0.17 | 0.21 | 0.22 | 0.20 |
| | Cheragh-Veys | 0.14 | 0.07 | 0.03 | 0.03 | 0.03 | 0.05 | 0.09 | 0.14 | 0.20 | 0.28 | 0.28 | 0.26 |
| | Markhuz | 0.09 | 0.04 | 0.02 | 0.02 | 0.03 | 0.06 | 0.11 | 0.15 | 0.20 | 0.21 | 0.20 | 0.15 |
| | Sonata | 0.09 | 0.04 | 0.03 | 0.03 | 0.05 | 0.08 | 0.12 | 0.16 | 0.19 | 0.20 | 0.18 | 0.14 |
| | Sarogh | 0.11 | 0.06 | 0.02 | 0.02 | 0.02 | 0.03 | 0.06 | 0.11 | 0.16 | 0.18 | 0.18 | 0.15 |

* A: Agricultural demand. E: Environmental requirement. D: Domestic demand.

## 4. Results and Discussion

Figure 5 shows the different hypothetical layouts implemented in R using *WRSS*. The R code for each conceptual model, shown in Figure 5, is accessible via CRAN repositories or other online HTML resources (https://rdrr.io/cran/WRSS/man/addObjectToArea.html (accessed on 18 August 2021)). The following sub-sections represent the application of *WRSS* in both design and operation of the Zerrine-rud water resources system.

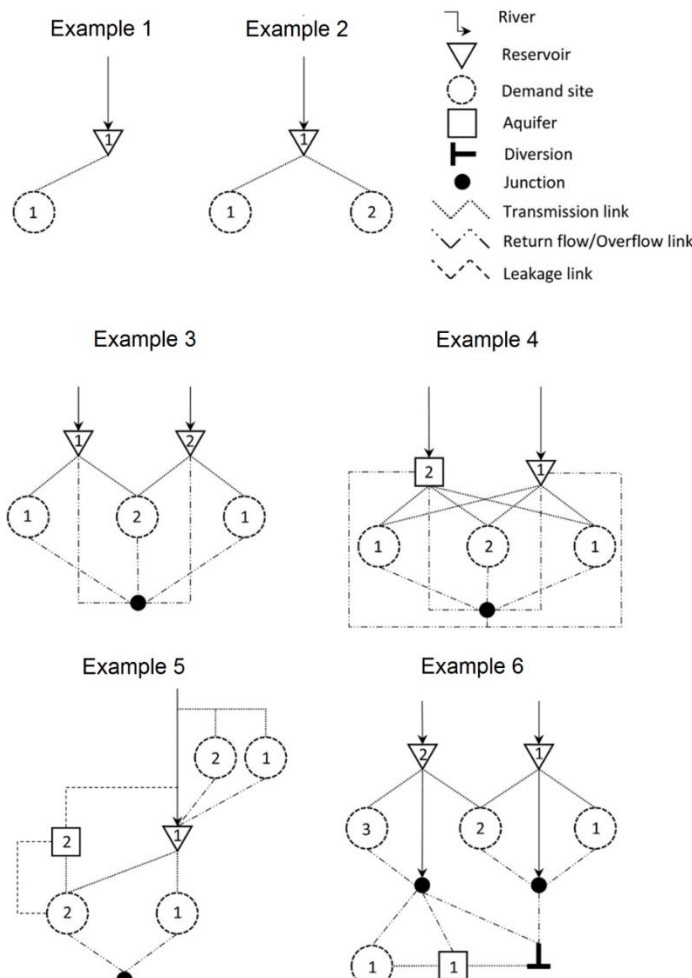

**Figure 5.** Conceptual water resources models implemented in R using *WRSS*. Object priorities are shown by values written over the objects (the smaller value the higher priority).

### 4.1. Capacity Design: Bukan Dam

For Bukan, the capacity redesigning *rippl* function is used to compute no-fail storage capacity using backward sequent-peak-algorithm for a given set of *target* and *discharge* flows. In the *rippl* method, a cumulative discharge time series ($Q_t$) was subtracted from a cumulative target ($D_t$) time series of ($S_t = \sum Q_t - \sum D_t$). Then, a backward search is conducted for every two subsequent peak and anti-peaks in $S_t$, and then they were compared and the largest ($S_t - S_{t-T}$) value was reported as the storage capacity required to meet the target time series. In Figure 6, after the computation of $S_t$, a backward search was performed to find the largest spike between every two subsequent peaks and anti-peaks. In the case of Bukan, in 10 years of monthly time series, the largest value was found between time steps 110 and 102.

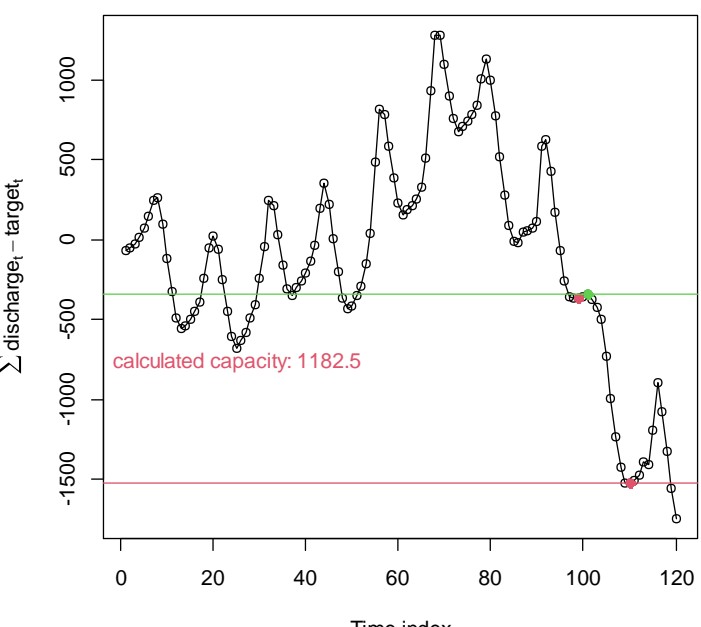

**Figure 6.** Diagram of Rippl's method for no-fail storage volume of the Bukan reservoir.

For annually averaged $1472 \times 10^6$ m$^3$ of inflow and $1645 \times 10^6$ m$^3$ of the target, the *rippl* function computes the reservoir capacity as $1181.2 \times 10^6$ m$^3$ and plots the results as shown in Figure 6.

In addition, a storage–yield relationship is analyzed by the functions and methods built into the package. To this end, let the yield–storage function be $\vartheta = \varnothing \left( Cap, \vec{\alpha} \right)$, where $\varnothing$ maps the capacity size, *Cap* and the other design parameter(s), $\vec{\alpha}$, such as the area of irrigable lands, to the RRV space, $\vartheta$. As a simple approach, the design parameter(s) domain(s) were discretized for the construction of the yield–storage relationship by evaluating all possible combinations of arguments. Accordingly, let the capacity of a dam be $Cap = \{S_1, S_2, \ldots, S_n\}$ and the Crop area be $Crop = \{A_1, A_2, \ldots, A_m\}$, RRV matrices, could be established from all pairs of $\left( Cap_i, Crop_j \right)$, where $i \leq n$, $j \leq m$. For the Bukan dam, let $Cap = \{500, 600, \ldots, 2000\} \times 10^6$ m$^3$ and $Crop = \{500, 600, \ldots, 2000\} \times 10^4$ m$^2$, and a pseudo-code, as shown in the Algorithm 2, generates a pairwise evaluation of the parameters over the given-above domain of the design parameters:

---

**Algorithm 2**

---

Initialize "*n*" design parameter(s) and discrete them within the search space

Make all possible combinations of design parameters, $\vec{M}$

**Loop**

    For each combination of design parameter(s), operate water resources feature(s).

    Evaluate RRV measures for every target(s).

    Terminate the loop if the criterion (number of iterations > number of combinations in *M*) is met

**End Loop**

---

*Cap_design* is an R implementation of the above-mentioned pseudo-code, which is able to plot the decision domain for each RRV measure. Figure 7a shows a graphical presentation of design variables with respect to risk-based indices. Based on the plots in Figure 7a, the domestic and industrial sectors function similarly with respect to design choices, however, the water requirement behaves differently, particularly for choices with higher capacities. In addition, a high dependency on the capacity size can be seen in all sectors for vulnerability and reliability indices. In contrast to the other sectors, the condition in the resiliency criterion is different, with multiple local optima and no general trend compared to the other measures. For domestic and agricultural sectors, the maximum

resiliency corresponds to a capacity around $700 \times 10^6$ m$^3$, where the decreasing gradient of vulnerability is becoming smoother along the capacity axis. However, to make the reservoir resilient to water supply requirements, the smallest capacity with the best resiliency is around $1200 \times 10^6$ m$^3$ with $900 \times 10^4$ m$^2$ of cropland area, which is 50% larger than the existing capacity size. Figure 7b presents SI and RSI measures calculated for the Bukan dam. Similar to Figure 7(a, b2, b3), domestic SI and agriculture SI, respectively, have similar trends, while Figure 7(b1) (water requirement SI) follows the water requirement resiliency surface in Figure 7a. To process SI aggregation, an AHP was conducted, and the weights of RSI were derived as 0.397, 0.332, and 0.270 for the environmental, domestic, and agricultural sectors, respectively. As a result, the RSI is calculated as presented in Figure 7(b4). Based on Figure 7(b4), the RSI maximum value is derived as 0.18, which corresponds to $1600 \times 10^6$ m$^3$ capacity and $900 \times 10^4$ m$^2$ cropland area. However, several local optima are suggesting smaller capacity values but with relatively lower RSIs. One appropriate candidate could be the $1200 \times 10^6$ m$^3$ capacity and $1000 \times 10^4$ m$^2$ cropland area with 0.13 RSI, which is quite close to the capacity size calculated by the ripple method ($1181.2 \times 10^6$ m$^3$). Another candidate design might be $600 \times 10^6$ m$^3$ with a crop area between $800 \times 10^4$ m$^2$ and $1200 \times 10^4$ m$^2$. According to the river flow regimes in the past decade, if developers had designed the dam with $150 \times 10^6$ m$^3$ smaller capacity, they would have supplied water requirement by almost the same rate of reliability as it does now. By contrast to the water requirement, Figure 7a signifies that the regulation of river flow for agricultural or domestic sectors requires a larger reservoir than the existing one. If the irrigation water and domestic water requirements are supplied by over 80% of reliability and the cropland area is the same as the current condition ($1000 \times 10^4$ m$^2$), a hypothetical reservoir size of $1000 \times 10^6$ m$^3$ is required to feed the croplands, (~$240 \times 10^6$ m$^3$ larger than the existing one).

*4.2. Large-Scale Simulation: Zerrine-Rud River Basin*

Features existing in the Zerrine-rud layout (see Figure 4b) were constructed using the functions presented in Table 2. Then, the basin object was simulated using the '*sim*' function on a computer with an Intel (R) Core (TM) i7-4790 CPU (4.00 GHz) CPU and 32 GB of installed memory (RAM). The simulation run time took 0.1218315 s on a 64-bit operating system, which is far faster than its rivals (e.g., WEAP with 5.73 s).

Figure 8 displays performance criteria derived by the *risk* method applied on the simulated basin object resulted from *sim* class. According to the chart, in parallel with the other studies reported in the literature [60,61], the computed reliability is significantly higher than resiliency based on SOP. For some targets, such as $U_1$, $U_2$, $A_1$, and $A_2$, an even larger difference can be viewed. Markhuz and Sarogh reservoirs outperformed with the highest reliability and resiliency and the least vulnerability. However, in addition to Sonata's vulnerable performance in E3 and A4 with the vulnerability indices of about 23 and 20, respectively, it has relatively lower reliability and resiliency than the others. Cheragh-Veys and Bukan conditions are quite similar with relatively the same performance in vulnerability and resiliency for domestic sectors ($U_1$ and $U_2$), while the Bukan dam has exceeded Cheragh-Veys in reliability and resiliency in the agricultural sector.

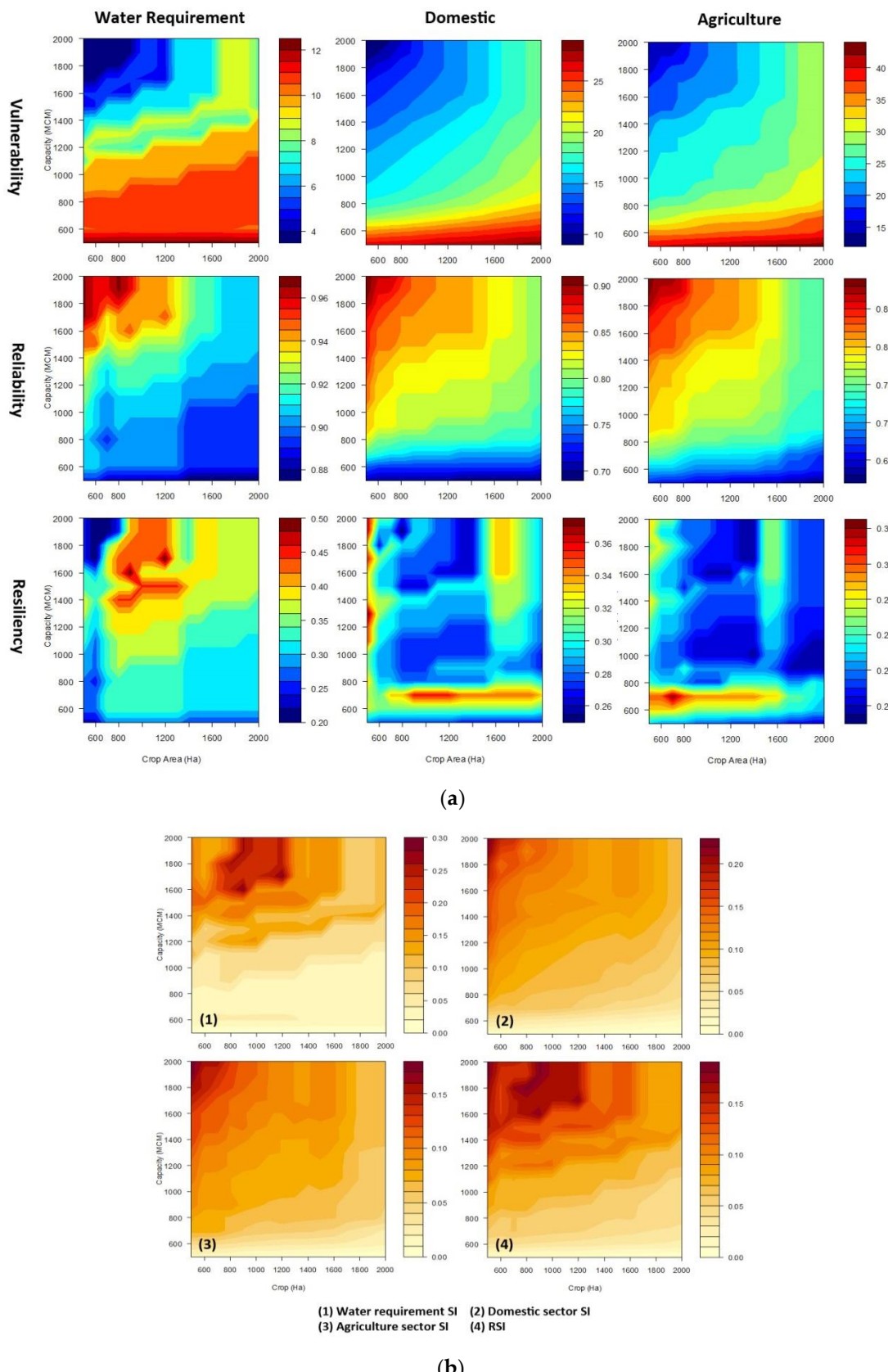

**Figure 7.** RRV and sustainability index-capacity-irrigated area relationships for Bukan reservoir. (**a**) RRV measures (**b**) 1 to 3 show sustainability indices (RS) for water requirement, domestic, and agricultural water use, respectivelym and 4 denotes the relative sustainability index (RSI).

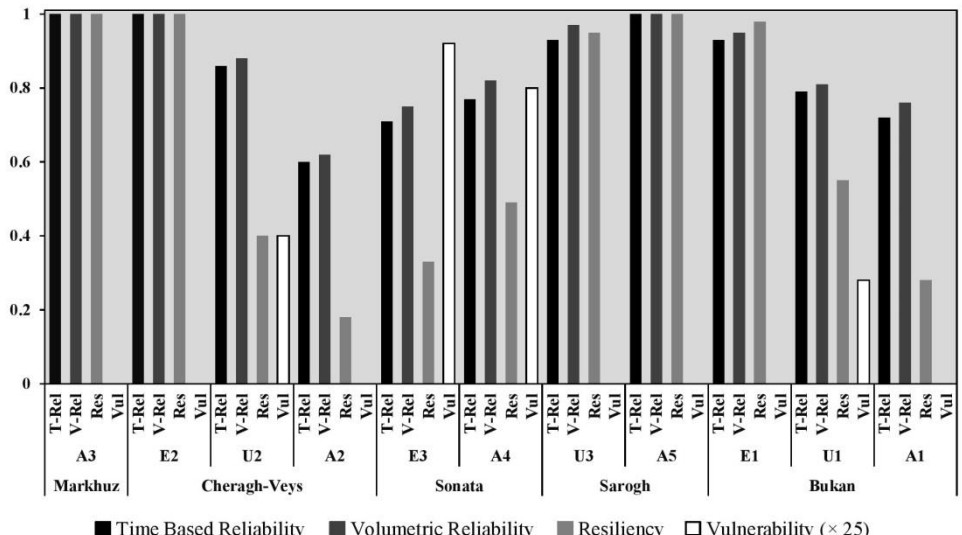

**Figure 8.** Risk-based performance criteria computed for Zerrine-rud demand sites.

Table 4 presents the performance criteria of the Bukan dam in supplying its target sites under single-unit and large-scale models. The table shows *WRSS* functionality in the enhancement of the operation of the Bukan dam by adding return-flow and spillage volumes from demand sites and reservoirs situated in its upstream. The annual average volumes contributing to the Zerrine River from upstream is $101.4 \times 10^6$ m$^3$, which is just below 10 percent of the current natural regime of the river. Under large-scale simulation, Bukan dam is perfectly resilient in supplying E1, while the resiliency criterion for the same site under isolated unit simulation is just over 0.3. By contrast to E1, the reliability and resiliency of U1 and A1 under large-scale simulation compared to the single unit has not been enhanced significantly, whereas the vulnerability has dropped notably by around 3.5 and 6.8, respectively.

**Table 4.** Risk-based criteria for the Bukan dam without and under the effect of upstream.

| Operation Type | Criteria | E1 | U1 | A1 |
|---|---|---|---|---|
| Single Unit Operation | Vulnerability | 10.758 | 17.122 | 26.837 |
| | Volumetric Reliability | 0.900 | 0.808 | 0.767 |
| | Time-based Reliability | 0.899 | 0.804 | 0.765 |
| | Resiliency | 0.333 | 0.304 | 0.250 |
| Large-scale Operation | Vulnerability | 0.137 | 13.870 | 20.000 |
| | Volumetric Reliability | 0.992 | 0.850 | 0.833 |
| | Time-based Reliability | 0.991 | 0.845 | 0.830 |
| | Resiliency | 1.000 | 0.333 | 0.300 |

Supplied volumes are displayed in Figure 9. The figure displays the reservoirs' monthly average releases throughout the simulation period superimposed by 95 percent release confidence and demand bar-chart. Excluding the Sarogh and Markhuz reservoirs, the performance of the other dams is not affected significantly by release quantities for the targets, specifically in the cropping season (May to October) when the demand for irrigation is high. In addition, the Sarogh and Markhuz dams have shown to be reliable in supplying both irrigation and domestic demands (Figure 9e). However, for the other reservoirs, despite saving water from non-cropping seasons, the reservoirs fail to cover the demands in the subsequent irrigation period. For the Sonata dam, this is even more apparent, where large quantities of demands (about 95%) are unsupplied, indicating a largely defined demand size for the Sonata reservoir and signifying the necessity for its justification.

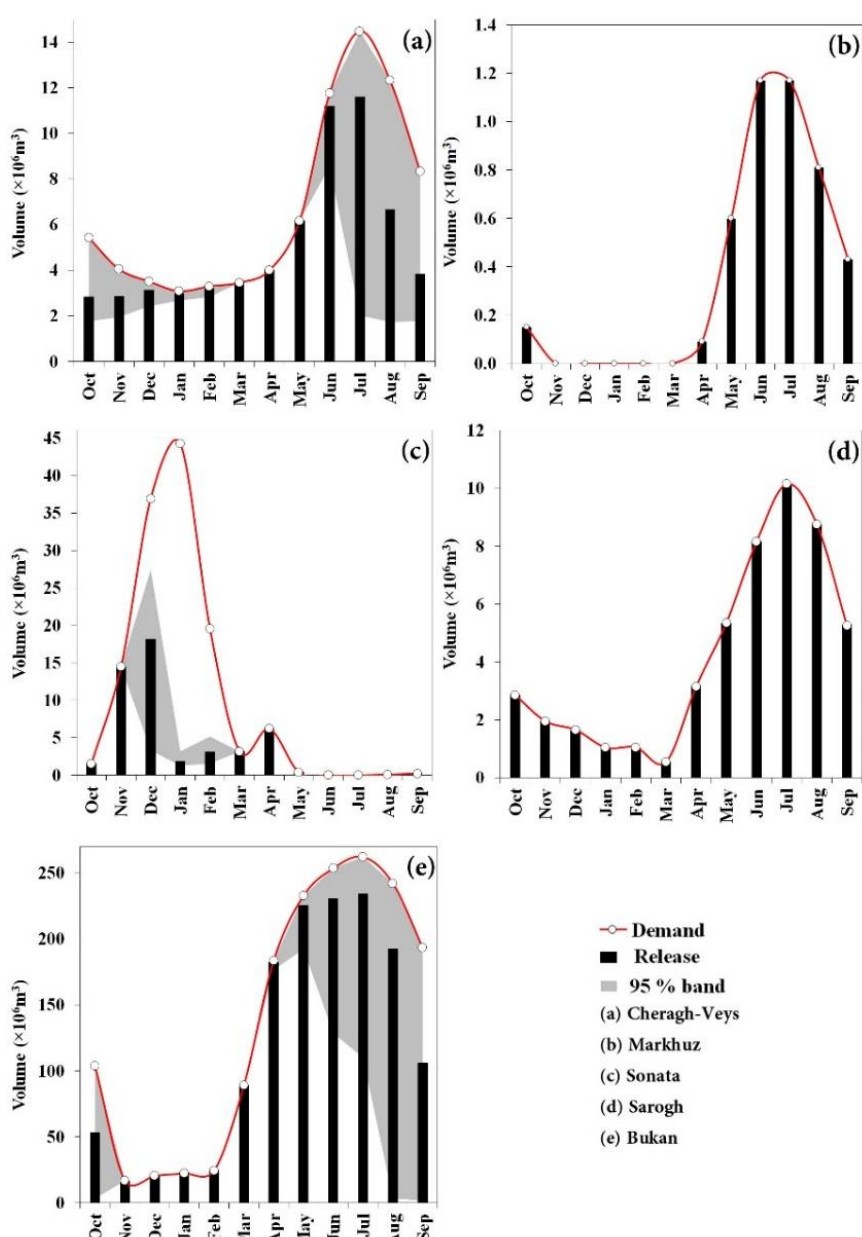

**Figure 9.** The results of monthly average simulation of the Zarrineh-Rud river basin.

## 5. Conclusions and Remark

The CRAN policies allow developers to upgrade and support packages, fix bugs, and extend software. Already, many capabilities have been added to the *WRSS* such as conjugate water resources operation, hydroelectric simulation, followed by fixing some bugs and errors.

Although there are many water resources simulation models, the *WRSS* implementation in R can be of a profound advantage to hydrologists and water resources engineers to address the impact of different hydro-climatic scenarios under different hard-work practices such as multiple structural works, e.g., dams, diversion, channels, etc. This is very important because the simulation of water system features in isolation does not take into account the impact of other components, consequently, not yielding precise and rational results.

The simulation of the Zerrine-rud basin reported the importance of object-oriented programming used in *WRSS*. Large-scale simulation of this basin supported the integration

of basin components through the implemented hydrological mechanisms, adding up 9.66% to the Bukan inflow through a contribution of 0.76% charged by upstream seepages and 8.90% coming from upstream spillages. Although the seepage contribution is around 7.9% of the added volume, it has been constantly contributed throughout the whole simulation period and it was the main factor in reducing the vulnerability index for one of the environmental water requirement demands (E1). This is because the E1 time series has an almost uniform distribution and a uniformly distributed supply, e.g., seepage, can significantly affect its reliability-resiliency-vulnerability metrics enhancement. However, even if the spillage volumes coming from upstream may yield large values, they happen occasionally with the lower possibility of their regulation, which consequently leads to less contribution to the enhancement of RRV metrics.

Utilizing *WRSS*, engineers and water resource scientists are able to assess, design, and operate water resources systems within the R environment. It is shown that the *WRSS* can have diverse applications in hydrologic analysis and large-scale basin modelling. To this end, a number of main advantages of *WRSS* can be summarized as below:

1. *WRSS* is an object-oriented R package supporting the simulation of large-scale supply water resources systems with complex layouts. The particular coding system devised for *WRSS* makes it possible to construct as many features as possible and include them in the simulation process.
2. The *WRSS* package can detect supply and allocation priorities for both water resources objects and demand nodes which have not been introduced in other R packages as well as many other open-source tools. Prioritization can be applied to demand features using shared or individual resources with any arbitrary priority. Furthermore, this is applicable for resource nodes where there are preferences in operation priorities.
3. *WRSS* provides constructors of objects in the basin rather than reservoirs, e.g., diversions, aquifers, etc., with the capability of interacting through mechanisms such as leakage, seepage, etc., which have been not available in other R packages. Additionally, the results demonstrate the importance of these mechanisms. Unless these mechanisms contribute to a small portion of the flow of the drainage network, they have significant impacts on the performance criteria.
4. *WRSS* is freely available, and R users can have the advantages of using the R's world of options. All of these possibilities could be used in the combination with *WRSS* objects to synergize its application in water resources modelling such as making coupled models under R platform.

**Software Availability**
Name of software: *WRSS*
Version: 3.0
Developers: Arabzadeh, R., et al.
Maintainer: Arabzadeh, R. <rezgararabzadeh@ut.ac.ir >
Year first available: 2017
License: GPL-3
Available from: Comprehensive R Archive Network (CRAN)
https://cran.r-project.org/package=WRSS

**Author Contributions:** Conceptualization, R.A. and P.A.; methodology, R.A., S.N. and R.S.; software, R.A.; validation, S.H., M.H. and W.R.; formal analysis, R.A. and P.A.; investigation, R.S. and R.A.; resources, R.A.; data curation, R.A. and P.A.; writing—original draft preparation, R.A., P.A., S.H., M.H. and S.N.; writing—review and editing, W.R. and R.S.; visualization, R.A.; supervision, R.S., W.R. and S.N.; project administration, R.S. and R.A.; funding acquisition, R.S. All authors have read and agreed to the published version of the manuscript.

**Funding:** This research received no external funding.

**Institutional Review Board Statement:** Not applicable.

**Informed Consent Statement:** Not applicable.

**Data Availability Statement:** Some or all data, models, or code generated or used during the study are available in a repository online at: https://github.com/rarabzad/WRSS. (accessed on 18 October 2021).

**Acknowledgments:** The authors wish to appreciate anonymous reviewers, Aram Jalali-Bouraban, and Samaneh Seifollahi-Aghmiuni for their comments on the paper structure and the manuscript proof reading.

**Conflicts of Interest:** The authors declare no conflict of interest.

## Annotations

| | |
|---|---|
| ***i:j:k*** | Feature index |
| | Outflow ($\times 10^6$ m$^3$) |
| *Reservoir*: | |
| $S_t^k$ | Storage ($\times 10^6$ m$^3$) |
| $Q_{t,k}^i$ | Inflow ($\times 10^6$ m$^3$) |
| $Sp_t^k$ | Spillage ($\times 10^6$ m$^3$) |
| $\overline{EV}_t^k$ | Average evaporation ($\times 10^6$ m$^3$) |
| $Re_{t,d}^k$ | Release for the $d^{th}$ demand node ($\times 10^6$ m$^3$) |
| $S_{min}^k$ | Dead storage ($\times 10^6$ m$^3$) |
| $S_{max}^k$ | Capacity ($\times 10^6$ m$^3$) |
| $A_t^k$ | Lake area ($\times 10^4$ m$^2$) |
| $E_t^k$ | Evaporation depth (m) |
| $Se_t^k$ | Seepage ($\times 10^6$ m$^3$) |
| $\omega^k$ | Seepage fraction [0,1] |
| *Aifer*: | |
| $V^k$ | Aquifer volume ($\times 10^6$ m$^3$) |
| $\varphi^k$ | Specific yield [0,1] |
| *Dand*: | |
| $D_t^k$ | Demand ($\times 10^6$ m$^3$) |
| $\delta_t^k$ | Effective supplied water ($\times 10^6$ m$^3$) |
| $Ex_t^k$ | Excess supplied water ($\times 10^6$ m$^3$) |
| $RV_t^k$ | Return flow ($\times 10^6$ m$^3$) |
| $RF^k$ | Return flow fraction [0,1] |
| *Diversion*: | |
| $Dv_t^k$ | Diverted water ($\times 10^6$ m$^3$) |
| $Cap^k$ | diversion capacity (m$^3$/s) |
| *Per plant*: | |
| $P_t$ | Generated energy (Watt) |
| $\rho$ | Water density (~1000 Kg/m$^3$) |
| $g$ | The gravity acceleration (~9.806 m/s$^2$) |
| $H_t$ | Gross head (m) |
| $\varphi_t$ | Power plant efficiency [0,1] |
| $hf_t$ | Total head losses (m) |
| $H_t^{(1)}$ and $H_t^{(2)}$ | Headwater between two subsequent time step |
| $h_t^{tail}$ | Head of tailwater (m) |
| $TAE$ | Turbine axis elevation (m) |
| $h_t^{tw}$ | Tailwater head in the river (m) |
| $hf_t^T$ | Turbine losses (m) |
| $hf_t^P$ | Penstock losses (m) |
| $L, D$ and $C$ | Length, Diameter, and Hazen–Williams Coefficient [62] |
| $\varnothing$ | Interpolator function of discharge-efficiency-table |
| $P_{Installed}$ | Installed capacity (Watts) |
| $DH$ and $DQ$ | Ranges of design head (m) and design flow rate (m$^3$/s) of the turbine |

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
