# Peer review of "WRSS: An Object-Oriented R Package for Large-Scale Water Resources Operation"

_water, doi:10.3390/w13213037_

Round 1

Reviewer 1 Report

Title: WRSS: An Object-Oriented R Package for Large-Scale Water Resources Operation

By: Rezgar Arabzadeh, Parisa Aberi, Sina Hesarkazzazi, Mohsen Hajibabaei, Wolfgang Rauch, Saman Nikmehr, and Robert Sitzenfrei

General comments

This manuscript was trying to use R package “WRSS” for analyzing the water resources planning. The core governing equation is mass conservation and optimal computation. The reviewer deemed this is the topic that the readers of “Water” would be interested in, although some flaws and question of Reviewer should be satisfied after a minor revision.

  1. the authors claimed that the application of R package in large scale analysis was the feature of this study. However, the reviewer did not agree with this statement. For instance, the authors did not define how large of a catchment could be a large-scale region. Then, some papers, had used R package as a tool to analyze global water resources (Turner and Galelli, 2016). The terminology “large-scale“ is ambiguous in this paper. Is that possible to add comprehensive literature review on this topic.
  2. about the governing equations. The authors claimed that the used package was applicable to treat seepage and pressure dominated pipes. However, the offered governing equations were only mass conservation and optimal algorithm, while the water flows in ground and pipes are momentum related problem. Since the governing equation did not related to the water momentum, how the computation could be valid in this paper. The authors shall supplement a reasonable explanation. If the momentum effects could be neglected in large-scale because of general surface flow dominance, then how was the applicability of this package in a small-scale area?
  3. Since the package had been developed for several years, some equation used in this paper should be documented in some files. The reviewer thinks the citations should be well marked in this paper. For example, the equations or figures should be added the sources beside them.

Detailed comments

  1. Table 1, some words are blocked, and how these levels were determined, from literatures or computed by authors?
  2. L84-85, please add citations about this case.
  3. L88-90, the paragraph is not clear enough to understand what the goals are in this article. The reviewer suggests that the authors to directly point out the main contribution or target of the paper rather than just writing the functions of WRSS.
  4. L224, the ramification of water use is only listed by domestic and agricultural. Does the use type influence the results? If it does, how come the industrial use was not included? If it does not, why did the authors list this example? Can different types be lumped as calculation?
  5. Figure 5, some example numbers are not correct. Moreover, the reviewer noted that this figure is identical to the example in the “Package WRSS” (https://cran.r-project.org/web/packages/WRSS/WRSS.pdf). If the figure is from other references, the authors are suggested to cite them beside the figure.
  6. L381-385, a little vague, how the WRSS mainly cope with in the algorithm.

References:

Turner, S. W., & Galelli, S. (2016). Water supply sensitivity to climate change: An R package for implementing reservoir storage analysis in global and regional impact studies. Environmental Modelling & Software76, 13-19.

Author Response

Dear Reviewer,

Regards,

Saman Nikmehr

Reviewer 2 Report

The manuscript presents a newly developed software package to simulate and model water resources system. The model is developed on an open-source platform with object-oriented programming. This model has a wide range of capabilities compared to other similar models, and it will be definitely of interest to the hydrology and water resources community. The paper is well written and easy to understand. I have only a few minor comments (typos)

  1. Table 1: Some texts in the Capabilities section were cut off. Please check.
  2. Line 136: I think it is “excess,” not exess.
  3. Line 146: Objects prioritization – capitalization
  4. Line 247: degree symbol
  5. Line 298, 299 – it should be 10^6

Author Response

Dear Reviewer,

Regards,

Saman Nikmehr

Reviewer 3 Report

I think the paper is of interest for Water's readers. The paper is well written and, after some improvements, it can be reconsidered for publication.

Here some point to address.

  • Lines 52-53. Please rephrase. Maybe better “For water resource evaluation purposes, there currently is a limited number of simulators, most of which are commercial or non-open source”
  • Lines 95-100: Authors describe the use of R in the context of Water Resources Management. I think a more comprehensive review of existing softwares is valuable, not only in R (which is definitely more suited for handling data), but also, for instance, in Python (e.g. Difonzo et al 2021 https://doi.org/10.1007/s11269-021-02850-2, Tomlinson et al 2020 https://doi.org/10.1016/j.envsoft.2020.104635, Dogan et al 2018 https://doi.org/10.1016/j.envsoft.2018.07.002)
  • Please check formatting of Table 1.
  • Equation (1). Please specify correctly the meaning of terms in the equation S, Q and O, and how it stems from “the core equation for water resources components”. O is defined only some lines later, I think it is better to place its definition just after eq (1).
  • Same argument for Eqs (3). Please define properly and clearly each term representing the outflow.
  • Line 135. Recalling a formula not yet stated is often improper
  • Line 136. “Exess”
  • Page 6 of 23: “Populate a by row reference matrix code whose columns show the features and rows are attributes of the features as follows”. I miss the sense of this sentence: please rephrase
  • I appreciate the choice of linking the pdf file with the description of packages.
  • Referring to table 2, I have some remarks. First: should the second column be labelled as Method? Second: Third column is objective, that is - as far as I can understand - output of the method. Could you please characterize more deeply what that is? I think that the output of those constructors are objects with properties: what are these properties? Could you please add some more details and explanations about them?

Author Response

Dear Reviewer,

Regards,

Saman Nikmehr

Reviewer 4 Report

Dear Authors

Unfortunately, the manuscript lacks new material and is very weak scientifically.
The manuscript is thus unacceptable.
Please pay attention to the text comments of the article.
Thanks

Author Response

Dear Reviewer,

Regards,

Saman Nikmehr

Round 2

Reviewer 3 Report

Accept

Reviewer 4 Report

Dear Author

Unfortunately, the manuscript does not present a new discussion in water engineering.
Therefore, in my opinion, the manuscript is not as good as the journal Water and I suggest rejecting it again.